# Remote assessment of pelvic kinematics during single leg squat using smartphone sensors: Between-day reliability and identification of acute changes in motor performance

**Valter Devecchi, Michelle Saunders, Sajni Galaiya, Millie Shaw, Alessio Gallina**  *

School of Sport, Exercise and Rehabilitation Sciences, University of Birmingham, Birmingham, United Kingdom

* a.gallina@bham.ac.uk

## Abstract

The biomechanical assessment of pelvic kinematics during a single leg squat (SLS) commonly relies on expensive equipment, which precludes its wider implementation in ecological settings. Smartphone sensors could represent an effective solution to objectively quantify pelvic kinematics remotely, but their measure properties need to be evaluated before advocating their use in practice. This study aimed to assess whether measures of pelvic kinematics collected remotely using smartphones during SLS are repeatable between days, and if changes in pelvic kinematics can be identified during an endurance task. Thirty-three healthy young adults were tested remotely on two different days using their own smartphones placed on the lumbosacral region. Pelvic orientation and acceleration were collected during three sets of seven SLS and an endurance task of twenty consecutive SLS. The intersession reliability was assessed using Intraclass Correlation Coefficient ($ICC_{2,k}$), Standard Error of Measurement, and Minimal Detectable Change. T-tests were used to identify pelvic kinematics changes during the endurance task and to assess between-day bias. Measures of pelvic orientation and frequency features of the acceleration signals showed good to excellent reliability (multiple $ICC_{2,k} \geq 0.79$), and a shift of the power spectrum to lower frequencies on the second day (multiple $p<0.05$). The endurance task resulted in larger contralateral pelvic drop and rotation (multiple $p<0.05$) and increased spectral entropy (multiple $p<0.05$). Our findings demonstrate that reliable measures of pelvic kinematics can be obtained remotely using participants' smartphones during SLS. Smartphone sensors can also identify changes in motor control, such as contralateral pelvic drop during an endurance task.

## Introduction

Single leg squat (SLS) is an exercise commonly used to assess neuromuscular control strategies [1, 2]. For instance, contralateral pelvic drop during single leg squat has been associated to hip

**Data Availability Statement:** All relevant data are within the paper and its Supporting Information files.

**Funding:** The author(s) received no specific funding for this work.

**Competing interests:** The authors have declared that no competing interests exist.

muscle weakness or altered muscle recruitment [3], and is commonly observed in individuals with musculoskeletal conditions such as patellofemoral pain and lower limb tendinopathies [4, 5]. Single leg squats are often prescribed as exercise intervention in individuals with lower limb musculoskeletal disorders such as patellofemoral pain [1], focusing on the specific motor impairments identified during SLS [6, 7].

Traditionally, objective measures of pelvic kinematics have been obtained using optoelectronic systems. However, the use of this technology in clinical practice is largely precluded because these systems are expensive and time-consuming, and high technical skills are required. Thus, clinicians often rely on subjective observations, and summarise the overall quality of movement patterns with a rating scale [8]. The inter-rater reliability of this approach is moderate [8], and a recent longitudinal study showed that the information obtained from visual assessment of the SLS is not useful to predict future lower limb injuries [9]. While visual observation has several advantages, including being no-cost and widely available, key biomechanical features of movement during SLS such as pelvic orientation and balance [5, 10] may require objective measures. Over the last decade, wearable technologies such as inertial measurement units (IMUs) have been proposed as an easier and cheaper solution to obtain reliable and valid objective measures for the assessment of movement patterns [11, 12]. Although the use of wearables has been advocated for objective movement assessment, both in the clinic and remotely, some issues still remain since the device and related software are often too expensive, and IMUs are currently not widely available to clinics or individuals.

Since smartphones are equipped with inertial sensors similar to those in IMUs, they may be a zero-cost, widely-available technology that may further facilitate the objective assessment of kinematic parameters in practice. There is evidence to support the use of smartphone for movement analysis given the good reliability and validity shown in the assessment of range of motion and static balance [13, 14]. However, the extraction of biomechanical features during dynamic tasks such as the single leg squat has received little attention. In addition, in most of the studies smartphone-based assessments were conducted in the laboratory, resulting in a low ecological validity when one of the largest advantages of this approach is the potential to use smartphones to measure movement objectively in clinical setting and remotely. A necessary step for the implementation of these technologies in practice is therefore to assess whether smartphones can be used to obtain objective motion data remotely, which could be useful to track changes in kinematics over time as well as patients' adherence to exercise prescription.

In this study we aimed to assess whether pelvic orientation measures collected remotely using a smartphone during a single leg squat are repeatable between days. We hypothesized that 3-dimensional pelvic kinematics, as well as frequency descriptors calculated from the acceleration signals, would demonstrate between-day repeatability comparable to that reported for laboratory techniques [15]. Furthermore, we assessed whether smartphone sensors can identify changes in pelvic kinematics during an endurance task. We hypothesized that, similarly to what reported in laboratory studies using optoelectronic systems, repeated single leg squats would result in larger contralateral pelvic drop and rotation [16].

## Methods

### Participants

Participants were recruited from the general community of Birmingham (UK) in March and April 2021. The sample size was calculated using an online application [17] and the following parameters: alpha = 0.05, power = 80%, minimum acceptable ICC = 0.7, expected ICC = 0.9 (based on other reliability studies of smartphone sensors [13]). The calculation resulted in a minimum sample size of 23 participants. However, to account for a potential larger-than-usual

dropout rate due to the remote nature of the study, we increased the number of participants to 33. Participants were eligible if they were between 18 and 65 years old and if they had access to a laptop and smartphone with an internet connection. Eligibility was verified through online questionnaires, and the following exclusion criteria were applied: current lower limb or back pain; lower limb injury in the previous 12 months, history of lower limb or spine surgery; neurological conditions or other major pathologies that limit movement or balance; pregnant at the time of recruitment or in the previous 12 months; inability to verbally communicate in English. All participants provided written informed consent digitally before participation in accordance with the Declaration of Helsinki. Demographic data and identification of participants were accessible only by investigators using RedCap (Research Electronic Data Capture); identification codes were used otherwise. The local ethics committee of the School of Sport, Exercise and Rehabilitation of the University of Birmingham approved this study (MCR180121-01).

## Procedure

Participants attended two sessions in separate days. Both sessions were conducted remotely, and participants were asked to be in the same environment of their preference. The performance of the exercise was supervised by final-year physiotherapy students via video-conference call (Zoom). Each participant was supervised by the same researcher in the two sessions, and, overall, four physiotherapy students were involved in the data collection. Age, anthropometrics and leg dominance (which leg they would use to kick a ball) were self-reported on an online questionnaire.

Participants performed a short warm-up of choice and familiarization with the task before the start of the data collection. To standardize the depth of the squat, participants were asked to find a surface as high as the middle of their thigh and squat until their gluteus lightly touched the top of this reference surface. Participants were asked to stand so that their heel was approximately 5 cm away from the reference surface. The pace of the exercise was standardized using a metronome. Participants were asked to squat down in 2 s, stay for 2 s, stand up in 2 s, and stay for 2 s. In each session (day 1 and day 2), participants performed 3 sets of 7 single leg squats both on their dominant and non-dominant legs in randomized order. The last task of each session (endurance task) was to perform a set of 20 consecutive SLS on their dominant and non-dominant leg (randomized).

## Data collection

Data were collected using Matlab Mobile App (The MathWorks, Natick, Massachusetts) installed on the participant's own smartphone (Android or iOS) and shared with the investigator through Matlab Drive. The data was collected using the Acceleration and Orientation sensors with sampling frequency set to 100 Hz.

Participants started the data acquisition and then placed their smartphone on the skin on top of their sacrum, using the elastic band of their shorts to keep it in place during the data collection. The smartphone was always placed horizontally and parallel to the base of the sacrum (Fig 1), and the investigator checked with the participant that the phone was in the correct position before each task. Studies adopting a single IMU during the SLS and other functional tasks identified the sacrum as the most suitable region because relevant time-domain and frequency-domain features can be extracted for the assessment of pelvic orientation and balance control [14, 18]. Participants were asked to interrupt the exercise at any point it they felt that the phone had moved or was about to fall. Five seconds of quiet standing with both feet on the

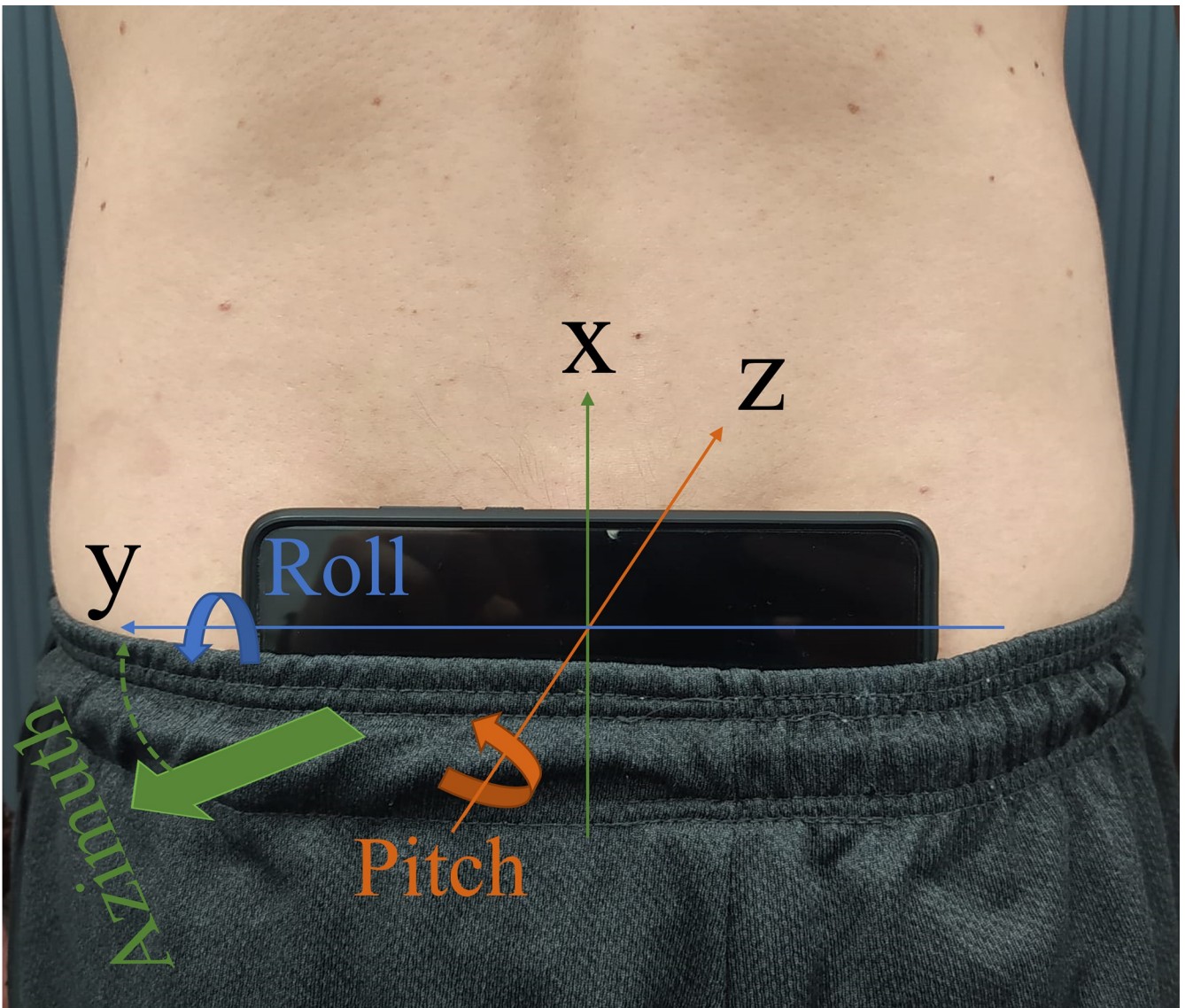

**Fig 1. Example of smartphone placement on the lumbosacral region, secured by the elastic band of the shorts.** Linear accelerations and orientation angles were obtained in reference to the three orthogonal axes X, Y and Z.

ground were performed before and after the execution of the task to obtain the reference orientation of the smartphone.

Fig 1 summarises the phone position with the axes of measurement, and the extracted signals with corresponding information are reported in Table 1.

## Data processing

All data were analysed using MATLAB R2021a (MathWorks). For all the sets with 7 repetitions, the first and last SLS repetition were disregarded to limit the potential effect of learning and fatigue, leaving 5 repetitions for each set. To limit the effect of the gravity component and high-frequency noise, the acceleration time-series on the three axes were bandpass filtered with cut-off frequencies at 0.2 and 20 Hz (Butterworth filter 4th order) [19, 20]. Then, the total

**Table 1. Signals and extracted features.** Note that the direction of the acceleration refers its main component; 'vertical' is not to be intended as 'exclusively vertical' acceleration, but as 'predominantly vertical'.

| Feature | Sensor | Description |
|---|---|---|
| *Time-domain signals and extracted features* | | |
| $a_x$ | Acceleration | Vertical linear acceleration along the x-axis |
| $a_y$ | Acceleration | Mediolateral linear acceleration along the y-axis |
| $a_z$ | Acceleration | Anteroposterior linear acceleration along the z-axis |
| Tilt | Orientation | Forward and backward pelvic tilt (sagittal plane); movement around the y-axis. A positive value identifies a forward tilt. |
| Obliquity | Orientation | Pelvic obliquity (frontal plane); movement around the z-axis. A negative value identifies a drop of the contralateral side (i.e., pelvic drop). |
| Rotation | Orientation | Pelvic rotation (transverse plane); movement around the x-axis. A negative value identifies a rotation toward the contralateral side (i.e., counter-clockwise when standing on the right leg). |
| *Frequency-domain features* | | |
| F50 | Acceleration | Median frequency; frequency below which 50% of total signal power is present. |
| F95 | Acceleration | Frequency that defines 95% of the power of the signal |
| SpE | Acceleration | Spectral Entropy of acceleration (unitless). |

acceleration was calculated from the three filtered time series as the Euclidean norm of the high-pass filtered signals as

$$HFEN = \sqrt{a_x^2 + a_y^2 + a_z^2}$$

Where *HFEN* is the Euclidean norm of the high-pass filtered signals, and *a* is the acceleration measured on each of the three axes.

Time-domain features were extracted during the squat phases of the SLS from the orientation waveforms. To identify the phases of the SLS, the displacement of the smartphone was computed from HFEN by numerical double-integration since the movement mainly occurred on the vertical axis. For each repetition between the 2nd and 6th, pelvic orientation was estimated as the average orientation in a 1-second window centered on the steady phase of each single leg squat. Orientation estimates were normalized to the initial position by subtracting the offset recorded during double-leg quiet standing. For the endurance task, time-domain features were extracted at the beginning (first 5 repetitions) and at the end (last 5 repetitions) of the task.

Frequency-domain features including median frequency (F50), frequency below which 95% of total signal power is present (F95), and spectral entropy (SpE), were extracted from the power spectrum of HFEN. The SpE of HFEN was calculated as [21]:

$$SpE = \frac{-1}{\log_{10} M} \sum_{i=1}^{n} [PSDi \times \log_{10}(PSDi)]$$

Where *PSDi* was the normalized value of the power spectrum density of HFEN at the $i^{th}$ frequency bin, and *M* denoted the number of frequency bins.

Frequency-domain features were computed in the time window between the second and sixth SLS repetitions (including the squat-down, squatting, return to standing, and standing phases, duration of approximately 20s). For the endurance task, frequency-domain features were extracted and averaged at the beginning (first 5 repetitions) and at the end (last 5 repetitions) of the task. Feature assessment was conducted for the dominant and non-dominant side separately. To facilitate the interpretation of results and integrate right and left side together,

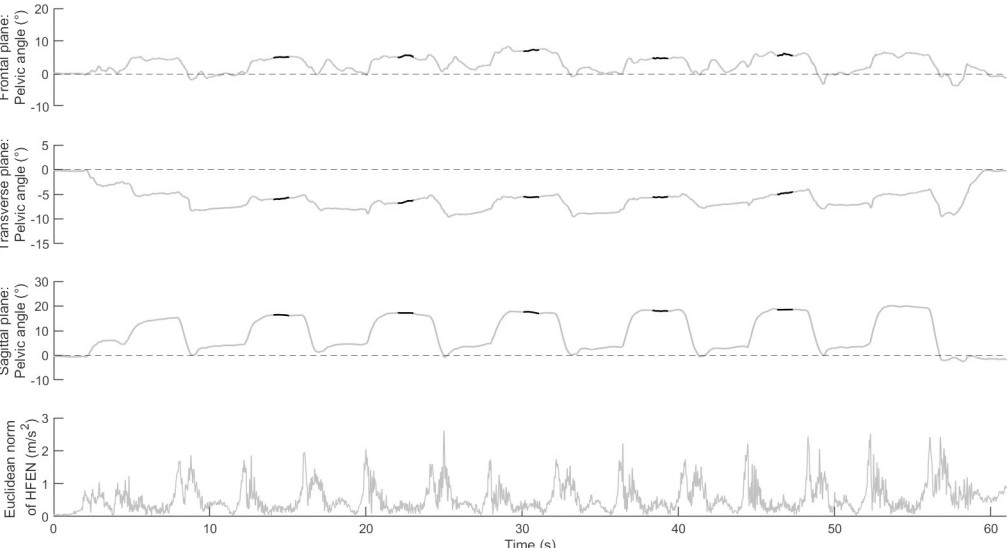

**Fig 2. Raw data from one participant during one set of single leg squat.** The orientation angles are illustrated for the frontal, transverse, and sagittal planes of movement. The thicker, black lines indicate the windows used to measure the orientation during the squat phase of the task (from the second to the sixth repetitions). Bottom: Euclidean norm of the high-pass filtered linear accelerations along the three axes.

the direction of movements was standardised (Fig 1) and summarised in Table 1. A summary of time- and frequency-domain features is presented in Table 1, and Fig 2 shows an example of raw data and processing.

## Statistical analysis

All statistical analyses were conducted in R software (R 4.1.0), separately for dominant and non-dominant leg. All data were visually inspected, and they were removed from the analyses if any misalignment was identified across repetitions (large change in baseline before and after the SLS). Descriptive statistics of the extracted features are presented as mean and standard deviation.

Between-day reliability was calculated for all extracted features using Interclass Correlation Coefficient (ICC [2,k]). The extracted feature represented the dependent variable, whereas participants and days were the random factors. The ICC for averaged measurements was considered since the features of interest were extracted and averaged across multiple repetitions. The ICC values were interpreted as follows: > 0.91 = Excellent, 0.90–0.75 = Good, 0.74–0.50 = Moderate, < 0.50 = Poor [22]. The standard error of measurement (SEM) was calculated as $SEM = SDpooled \times \sqrt{(1 - ICC)}$ [23]. The minimum detectable change (MDC) was then calculated as $MDC = SEM \times 1.96 \times \sqrt{2}$ [23]. Bland Altman plots were used to assess the limits of agreement between the two days and to identify any systematic bias, which was also tested with one sample T-test. Side differences in pelvic orientation and acceleration were tested using T-test or paired Wilcoxon test.

To detect changes in the features of interest during the endurance task, the average values at the beginning (first 5 repetitions) and end (last 5 repetitions) of the task were compared using paired t-test or Wilcoxon test depending on the data distribution assessed with Shapiro-Wilk test.

**Table 2. Demographic characteristics.**

| | *N = 33* |
|---|---|
| Gender (F / M) | 21 / 12 |
| Age; *mean(SD)* | 24.2 (4.0) |
| Height, cm; *mean(SD)* | 168.4 (9.7) |
| Weight, kg; *mean(SD)* | 68.0 (12.1) |
| Leg dominance (R/L) | 31 / 2 |

## Results

A total of 33 healthy participants were recruited and their demographic characteristics are presented in Table 2. Data from all participants were used for dominant and non-dominant leg comparisons; no between-leg differences were observed in pelvic orientation and frequency features (p>0.313), with the exception of a marginally higher value of F95 for the non-dominant leg (p = 0.036). Intersession reliability of time-domain features was calculated from 30 participants since 3 participants dropped out before session 2. One more participant was excluded from the assessment of reliability of frequency-domain features (N = 29) because no acceleration data were available on day 2 since the sensor was not activated in the experimental session. Pelvic kinematics during the endurance task were calculated from 30 participants, except for the frontal and transverse orientation on the dominant leg (n = 28) and transverse orientation on the non-dominant leg (n = 29). These data were excluded after visual inspection of the signals because of smartphone misalignment across repetitions (large change in baseline before and after the SLS).

### Between day reliability

Bland Altman plot are presented in Fig 3 for time-domain and frequency domain features extracted from the dominant leg. Similar limits of agreement are present between dominant and non-dominant leg. No proportional bias was identified for any feature of interest.

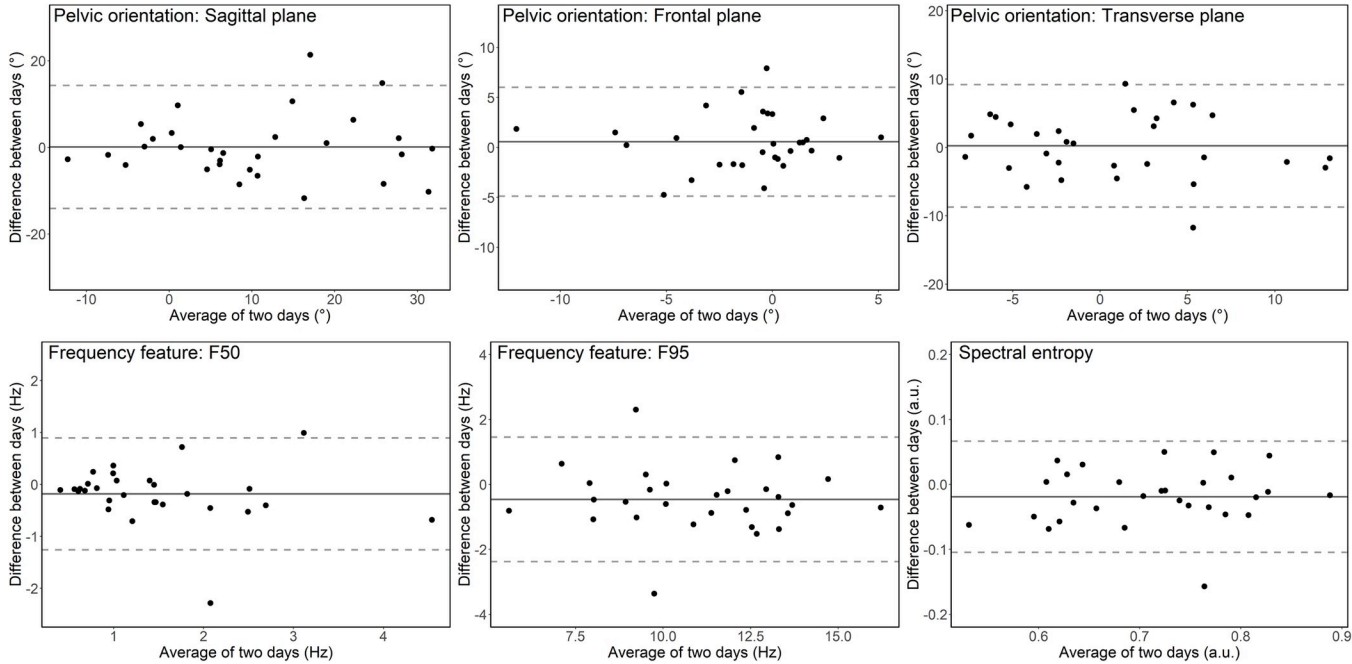

**Fig 3. Bland Altman plots showing the between-day reliability of pelvic orientation and frequency features (dominant side).**

**Table 3. Reliability of time-domain (N = 30) and frequency-domain (N = 29) features during a single leg squat.**

| Feature, side | ICC | SEM | MDC | Mean(SD) day1 | Mean(SD) day2 | p value |
|---|---|---|---|---|---|---|
| Tilt, Dom (˚) | 0.91 [0.82–0.96] | 3.72 | 10.31 | 10.31 (12.55) | 10.40 (12.85) | 0.944 |
| Obliquity, Dom (˚) | 0.84 [0.66–0.92] | 1.50 | 4.17 | -1.42 (3.86) | -0.85 (3.78) | 0.274 |
| Rotation, Dom (˚) | 0.85 [0.68–0.93] | 2.42 | 6.72 | 0.70 (6.43) | 0.93 (5.90) | 0.786 |
| Tilt, NDom (˚) | 0.89 [0.77–0.95] | 4.01 | 11.12 | 10.71 (12.53) | 9.56 (11.58) | 0.417 |
| Obliquity, NDom (˚) | 0.90 [0.80–0.95] | 1.52 | 4.20 | -0.78 (4.75) | -0.65 (4.94) | 0.800 |
| Rotation, NDom (˚) | 0.79 [0.56–0.90] | 2.15 | 5.95 | -0.15 (4.46) | -0.33 (4.86) | 0.805 |
| F50, Dom (Hz) | 0.90 [0.80–0.95] | 0.30 | 0.82 | 1.57 (1.00) | 1.38 (0.91) | 0.084 |
| F95, Dom (Hz) | 0.95 [0.90–0.98] | 0.54 | 1.51 | 11.24 (2.57) | 10.78 (2.50) | 0.017 |
| Entropy, Dom | 0.93 [0.85–0.97] | 0.02 | 0.07 | 0.72 (0.09) | 0.70 (0.09) | 0.027 |
| F50, NDom (Hz) | 0.84 [0.67–0.93] | 0.35 | 0.97 | 1.58 (0.98) | 1.27 (0.77) | 0.010 |
| F95, NDom (Hz) | 0.93 [0.86–0.97] | 0.59 | 1.62 | 11.59 (2.27) | 11.15 (2.26) | 0.034 |
| Entropy, NDom | 0.94 [0.87–0.97] | 0.02 | 0.06 | 0.72 (0.09) | 0.70 (0.08) | 0.021 |

Dom: dominant; NDom: non-dominant; Tilt: pelvic orientation on the sagittal plane; Obliquity: pelvic orientation on the frontal plane; Rotation: pelvic orientation on the transverse plane; F50: median frequency; F95: frequency that defines 95% of the power of the signal; Entropy: spectral entropy; ICC: Intraclass Correlation Coefficient; SEM: Standard Error of Measurement; MDC: Minimal Detectable Change; p value: statistical significance of the one sample t-test on the difference between days.

The ICC, SEM, MDC, and average values measured in Day1 and Day2 are presented in Table 3 for all time-domain and frequency-domain features and for both the dominant and the non-dominant leg. Interclass correlation coefficients showed good to excellent values for all investigated features (ICCs ranging from 0.79 to 0.95). Pelvic rotation on the non-dominant leg was the feature with the lowest ICC value (0.79). Intersession MDC values were similar between sides and larger for pelvic tilt (>10˚) compared to pelvic rotation (5–7˚) and pelvic drop (approximately 4˚). Pelvic orientation measures were similar between days (p≥0.274). Instead, all frequency-based features were lower on the second visit (p≤0.034), except F50 on the dominant leg (p = 0.084).

## Changes in motor performance during endurance task

Changes of time- and frequency-domain features during an endurance task (Table 4) were assessed by comparing the first 5 with the last 5 repetitions of the task (out of 20 repetitions). The endurance task resulted in larger contralateral pelvic drop when performing single leg squats both on the dominant (-1.34 ± 2.81˚, p = 0.018) and non-dominant leg (-1.96 ± 2.57˚, p < 0.001, Fig 4). Similarly, an increase of pelvic rotation toward the contralateral side for both the dominant (-1.97 ± 4.34˚, p = 0.024) and non-dominant leg (-3.90 ± 4.85˚, p < 0.001) was identified. No pelvic tilt differences were found, although a trend suggesting for an increase of forward tilt was observed when SLS was performed on the non-dominant leg (1.72 ±5.1˚, p = 0.064). Frequency domain features F50 and F95 did not change over the twenty repetitions (Fig 5); however, a small increase of entropy was observed for both dominant (0.025 ± 0.065, p = 0.040) and non-dominant leg (0.017 ± 0.038, p = 0.024).

## Discussion

This study aimed to evaluate the between-day reliability of pelvic kinematics during a SLS collected remotely using participants' smartphones. Pelvic orientation showed good to excellent between-day reliability, especially on the frontal and sagittal planes. Similarly, frequency descriptors of the pelvic acceleration showed good to excellent agreement but were

**Table 4. Changes of time-domain and frequency-domain features during an endurance task.**

| Feature, side | Pattern of movement | Mean(SD) first 5 | Mean(SD) last 5 | Mean diff [CI 95%] | p value |
|---|---|---|---|---|---|
| Tilt, Dom (°) | | 8.84(13.26) | 8.77(12.89) | -0.071[-1.45 to 1.31] | 0.685[#] |
| Obliquity, Dom (°) | Contralateral drop | -1.07(3.47) | -2.41(3.53) | -1.341[-2.43 to -0.25] | 0.018 |
| Rotation, Dom (°) | Rotation toward contralateral leg | 0.20(5.41) | -1.77(6.35) | -1.966[-3.65 to -0.28] | 0.024 |
| Tilt, NDom (°) | | 10.45(13.63) | 12.18(15.59) | 1.723[-0.18 to 3.62] | 0.064[#] |
| Obliquity, NDom (°) | Contralateral drop | -0.80(4.97) | -2.76(6.14) | -1.955[-2.92 to -1.00] | 0.000[#] |
| Rotation, NDom (°) | Rotation toward contralateral leg | -0.17(4.38) | -4.07(5.04) | -3.897[-5.74 to -2.05] | 0.000[#] |
| F50, Dom (Hz) | | 1.34(0.81) | 1.55(0.92) | 0.211[-0.08 to 0.50] | 0.147 |
| F95, NDom (Hz) | | 11.25(2.46) | 11.69(2.50) | 0.437[-0.02 to 0.89] | 0.059 |
| Entropy, Dom | | 0.73(0.08) | 0.75(0.08) | 0.025[0.001 to 0.049] | 0.040[#] |
| F50, NDom (Hz) | | 1.29(0.93) | 1.32(0.84) | 0.030[-0.22 to 0.27] | 0.626[#] |
| F95, NDom (Hz) | | 11.82(2.28) | 12.09(2.22) | 0.278[-0.32 to 0.87] | 0.584[#] |
| Entropy, NDom | | 0.72(0.08) | 0.73(0.08) | 0.017[0.002 to 0.031] | 0.024 |

Dom: dominant; NDom: non-dominant; Tilt: pelvic orientation on the sagittal plane; Obliquity: pelvic orientation on the frontal plane; Rotation: pelvic orientation on the transverse plane; F50: median frequency; F95: frequency that defines 95% of the power of the signal; Entropy: spectral entropy. [#]p value obtained from Wilcox Test, paired t-test otherwise

systematically lower in the second visit. When single leg squats were repeated in an endurance task, smartphone sensors were able to detect changes in pelvic drop and rotation. This study suggests that smartphone sensors are reliable and able to identify changes in pelvic orientation during single leg squat, supporting their use to assess and monitor performance of the single leg squat outside the laboratory.

In accordance with the high between-day reliability reported when testing static tasks and range of motion [13, 24, 25], our work adds to this evidence by showing that smartphone sensors provide reliable 3-dimensional pelvic orientation and acceleration data during a dynamic SLS task. A previous study conducted in the laboratory using an electromagnetic tracking system reported high reliability of pelvic orientation measures between days, with ICCs greater than 0.90 for all movement planes [15]. Our results revealed a similar trend since only

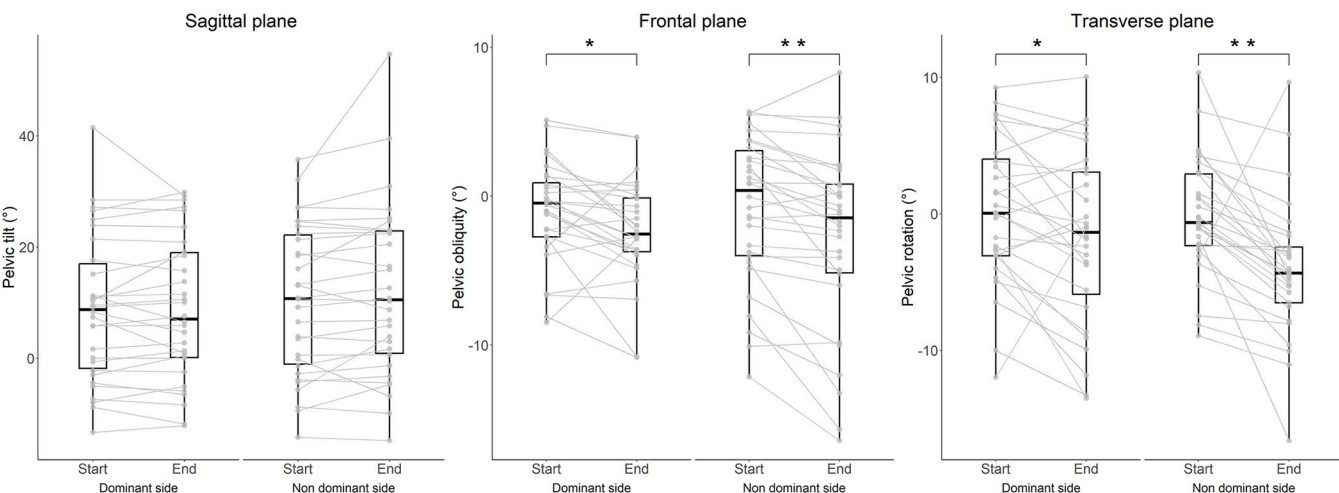

**Fig 4. Change of pelvic orientation between the start (first 5 repetitions) and end (last 5 repetitions) of single leg squat endurance task.** Data are shown for the dominant and non-dominant sides. *p<0.05 and ** p<0.01.

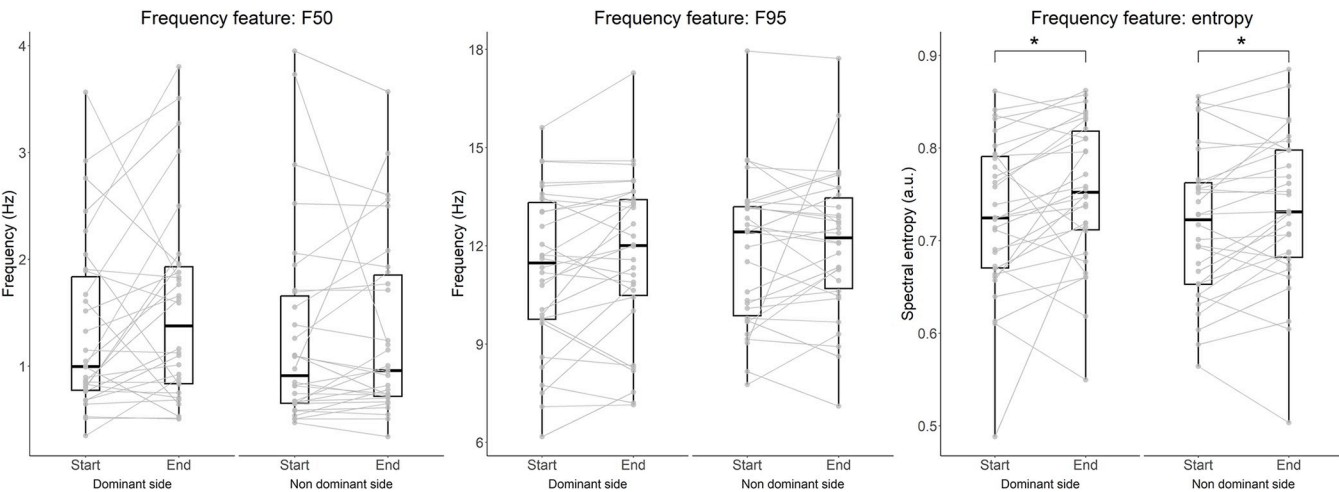

**Fig 5. Change of frequency features between the start (first 5 repetitions) and end (last 5 repetitions) of the single leg squat endurance task.** Data are shown for the dominant and non-dominant sides. *p<0.05.

orientation on the transverse plane (non-dominant side) showed an ICC lower than 0.80. Compared to pelvic orientation, frequency-domain features had higher agreement but were affected by a systematic bias. Measures collected during the second day showed a reduction of F50, F95 and entropy, demonstrating that the SLS was characterized by lower frequency components when it was performed the second time, which might indicate a learning effect related to better balance and/or smoothness of movement on the second day. Our findings of lower acceleration values in the second visit concur with the balance improvements reported after a single session of unilateral exercise in healthy individuals [26].

In line with a previous study using an electromagnetic tracking system [15], pelvic orientation on the sagittal plane showed the largest MDC whereas the frontal plane had the smallest MDC. We found an MDC of 4.2° on the frontal plane, 1° larger than that reported by Nakagawa et al. using an electromagnetic tracking system in the laboratory [15]. This MDC of pelvis orientation on the frontal plane is similar to the kinematic improvement observed after functional and strengthening interventions in clinical and healthy population ranging between 3.5 and 4° [27–29], which supports the possible usefulness of the proposed objective measure to monitor changes in pelvic kinematic over time, even during remote assessments. Differently from measures of pelvic orientation, clinical data on pelvic stability obtained from acceleration signals analysed in the time- or frequency domains are less available and less standardized, which makes comparisons across studies difficult. A main difference between this and most of the other studies is that measures of stability during SLS often focused on the knee (described as knee wobbling) rather than the pelvic region [30, 31]. Despite the gap in the literature which precludes direct comparisons across studies, the reduction of frequency-domain features reported on the second day is in accordance with other studies showing balance improvement after a single session [26], and indirectly supports the ability of smartphones to detect changes over time. Taken together, measures of between-day reliability are adequate to support the remote assessment of pelvic kinematics during a SLS using participants' smartphones.

To assess whether smartphone sensors can detect acute changes in kinematics, we evaluated whether pelvic orientation changes measured with smartphone sensors during an endurance task are similar to those observed when a fatigue task was performed in the laboratory. On the frontal and transverse plane, Weeks and colleagues observed increased contralateral in pelvic drop and rotation when individuals performed single leg squats after a fatiguing protocol [16].

We report similar kinematic changes when kinematics is measured using smartphone sensors remotely. The difference in magnitude and timing of these kinematic changes between studies, and the lack of significant differences in pelvic tilt in this study, is likely due to differences in how the task was performed. Based on several studies comparing clinical populations and the effect of interventions, a larger pelvic drop during a SLS is usually attributed to a deficit of hip abductor muscle strength or activation [3], which is a recommended target for exercise in the rehabilitation of individuals for patellofemoral pain [1]. Reinforced by the clinical meaning that the changes identified during the SLS endurance task have, the findings of this study support the potential use of smartphones to detect pelvic orientation changes during an endurance task as a part of remote clinical assessments. In contrast to pelvic orientation measures, frequency-domain features did not change during the endurance task, except for a small increase of entropy with the non-dominant side. Such differences in the pattern of changes between time- and frequency domain features suggest that they could assess different constructs, and that a SLS endurance task mainly influence pelvic orientation rather than balance or smoothness of movement.

The main strength of this study is its ecological validity. It was performed remotely under supervision in video-conferencing, and participants used their own smartphones to collect data, simulating a tele-rehabilitation session. This study demonstrates that highly reliable pelvic kinematic data can be collected even when participants place the smartphone on themselves.

This study is also affected by some limitations. The lower values of ICC obtained on the transverse plane might be influenced by the noise of the magnetometer sensors since we did not control for the presence of ferromagnetic objects close to the participants. However, the inspection of the data in the Bland-Altman plots revealed only two participants out of the limit of agreements and data showing any potential misalignments across repetitions (change in baseline before and after the SLS) were discarded. Also, we cannot ensure that all participants experienced the same amount of fatigue since we used an endurance task rather than a fatigue protocol because it was easier to standardise during a remote assessment. Finally, data on the construct validity of the adopted approach are required to further promote the use of smartphones for remote assessments.

## Conclusions

In conclusion, smartphone sensors provide reliable pelvic kinematic data during single leg squat in the time and frequency domain. This methodology also replicated some findings of studies performed in the laboratory, such as pelvic drop and rotation during an endurance task, and possibly balance improvement after a single session of training. These results show promise in the use of smartphone sensors to assess and monitor kinematics remotely, which may be useful in clinical practice.

## Supporting information

**S1 Data. Endurance task.**
(CSV)

**S2 Data. Reliability.**
(CSV)

## Author Contributions

**Conceptualization:** Alessio Gallina.

**Formal analysis:** Valter Devecchi.

**Investigation:** Valter Devecchi, Michelle Saunders, Sajni Galaiya, Millie Shaw.

**Supervision:** Alessio Gallina.

**Writing – original draft:** Valter Devecchi.

**Writing – review & editing:** Valter Devecchi, Michelle Saunders, Sajni Galaiya, Millie Shaw, Alessio Gallina.

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
