## [Decision Letter · Decision Letter 0]

15 Jun 2023

PONE-D-23-10433Remote assessment of pelvic kinematics during single leg squat using smartphone sensors: between-day reliability and identification of acute changes in motor performancePLOS ONE

Dear Dr. Gallina,

Thank you for submitting your manuscript to PLOS ONE. After careful consideration, we feel that it has merit but does not fully meet PLOS ONE’s publication criteria as it currently stands. Therefore, we invite you to submit a revised version of the manuscript that addresses the points raised during the review process.

ACADEMIC EDITOR: Dear Authors, one expert in the field reviewed your manuscript reporting some minor issues you should consider in the revision process.Please submit your revised manuscript by Jul 30 2023 11:59PM. If you will need more time than this to complete your revisions, please reply to this message or contact the journal office at plosone@plos.org. Please include the following items when submitting your revised manuscript:A rebuttal letter that responds to each point raised by the academic editor and reviewer(s). You should upload this letter as a separate file labeled 'Response to Reviewers'.A marked-up copy of your manuscript that highlights changes made to the original version. You should upload this as a separate file labeled 'Revised Manuscript with Track Changes'.An unmarked version of your revised paper without tracked changes. You should upload this as a separate file labeled 'Manuscript'.If applicable, we recommend that you deposit your laboratory protocols in protocols.io to enhance the reproducibility of your results. Protocols.io assigns your protocol its own identifier (DOI) so that it can be cited independently in the future. For instructions see: https://journals.plos.org/plosone/s/submission-guidelines#loc-laboratory-protocols. Additionally, PLOS ONE offers an option for publishing peer-reviewed Lab Protocol articles, which describe protocols hosted on protocols.io. Read more information on sharing protocols at https://plos.org/protocols?utm_medium=editorial-email&utm_source=authorletters&utm_campaign=protocols.

We look forward to receiving your revised manuscript.

Kind regards,

Emiliano Cè

Academic Editor

PLOS ONE

Journal Requirements:

Reviewers' comments:

Reviewer's Responses to Questions

**Comments to the Author**

1. Is the manuscript technically sound, and do the data support the conclusions?

Reviewer #1: Yes

2. Has the statistical analysis been performed appropriately and rigorously? 

Reviewer #1: Yes

3. Have the authors made all data underlying the findings in their manuscript fully available?

Reviewer #1: Yes

4. Is the manuscript presented in an intelligible fashion and written in standard English?

Reviewer #1: Yes

5. Review Comments to the Author

Reviewer #1: The feasibility of using smartphone sensors for biomechanical assessment of pelvic kinematics during a single leg squat (SLS) in ecological settings was investigated. Thirty-three healthy young adults participated remotely on two different days using their own smartphones placed on the lumbosacral region. Pelvic orientation and acceleration data were collected during SLS sets and an endurance task. The reliability of the measurements was assessed using Intraclass Correlation Coefficient (ICC2,k), Standard Error of Measurement, and Minimal Detectable Change. Results showed good to excellent reliability (ICC2,k ≥ 0.79) for pelvic orientation and frequency features of acceleration signals. Changes in pelvic kinematics during the endurance task were identified, including larger contralateral pelvic drop and rotation. These findings indicate that smartphones can provide reliable measurements of pelvic kinematics remotely during SLS, and also detect changes in motor control during an endurance task.

I have completed the review of the manuscript in question, and overall, I find the study to be well-written with an interesting practical application. Despite the use of "commercial" equipment, the methodological approach is robust. The results are clearly presented and comprehensive, as are the accompanying figures. The discussion is well-supported by the results.

I have only one point that I believe the authors should address: Was the calculation of the sample size performed? The term "convenient sample" is somewhat ambiguous, and it would be beneficial for the authors to clarify this aspect in the text, if such calculations were indeed conducted.

6. PLOS authors have the option to publish the peer review history of their article (what does this mean?). If published, this will include your full peer review and any attached files.

Reviewer #1: **Yes: **Christian Doria

---

## [Author Response · Author response to Decision Letter 0]

3 Jul 2023

Reviewer 1

The feasibility of using smartphone sensors for biomechanical assessment of pelvic kinematics during a single leg squat (SLS) in ecological settings was investigated. Thirty-three healthy young adults participated remotely on two different days using their own smartphones placed on the lumbosacral region. Pelvic orientation and acceleration data were collected during SLS sets and an endurance task. The reliability of the measurements was assessed using Intraclass Correlation Coefficient (ICC2,k), Standard Error of Measurement, and Minimal Detectable Change. Results showed good to excellent reliability (ICC2,k ≥ 0.79) for pelvic orientation and frequency features of acceleration signals. Changes in pelvic kinematics during the endurance task were identified, including larger contralateral pelvic drop and rotation. These findings indicate that smartphones can provide reliable measurements of pelvic kinematics remotely during SLS, and also detect changes in motor control during an endurance task.

I have completed the review of the manuscript in question, and overall, I find the study to be well-written with an interesting practical application. Despite the use of "commercial" equipment, the methodological approach is robust. The results are clearly presented and comprehensive, as are the accompanying figures. The discussion is well-supported by the results.

I have only one point that I believe the authors should address: Was the calculation of the sample size performed? The term "convenient sample" is somewhat ambiguous, and it would be beneficial for the authors to clarify this aspect in the text, if such calculations were indeed conducted. 

We thank the reviewer for the positive feedback about our study. 

Regarding the comment on the sample size, we expanded the section explaining that we adopted current statistical methods relying on the interclass correlation coefficients to identify the required sample size. 

Line 83: “The sample size was calculated using an online application (17) and the following parameters: alpha = 0.05, power = 80%, minimum acceptable ICC = 0.7, expected ICC = 0.9 (based on other reliability studies of smartphone sensors (13). The calculation resulted in a minimum sample size of 23 participants. However, to account for a potential larger-than-usual dropout rate due to the remote nature of the study, we increased the number of participants to 33.”

In addition to the revisions requested by the reviewer, we have also slightly revised the text for grammar and clarity (track-changed).

13. Keogh JWL, Cox A, Anderson S, Liew B, Olsen A, Schram B, et al. Reliability and validity of clinically accessible smartphone applications to measure joint range of motion: A systematic review. Müller J, editor. PLOS ONE. 2019 May 8;14(5):e0215806.

17. Arifin WN. A Web-based Sample Size Calculator for Reliability Studies. Educ Med J. 2018 Sep 28;10(3):67–76.

---

## [Editor Report · Decision Letter 1]

5 Jul 2023

Remote assessment of pelvic kinematics during single leg squat using smartphone sensors: between-day reliability and identification of acute changes in motor performance

PONE-D-23-10433R1

Dear Dr. Gallina,

We’re pleased to inform you that your manuscript has been judged scientifically suitable for publication and will be formally accepted for publication once it meets all outstanding technical requirements.

Kind regards,

Emiliano Cè

Academic Editor

PLOS ONE
---

## [Editor Report · Acceptance letter]

12 Jul 2023

PONE-D-23-10433R1 

Remote assessment of pelvic kinematics during single leg squat using smartphone sensors: between-day reliability and identification of acute changes in motor performance 

Dear Dr. Gallina:

I'm pleased to inform you that your manuscript has been deemed suitable for publication in PLOS ONE. Congratulations! Your manuscript is now with our production department. 

Kind regards, 

on behalf of

Prof. Emiliano Cè 

Academic Editor

PLOS ONE